# HIEGNet: A Heterogeneous Graph Neural Network Including the Immune Environment in Glomeruli Classification

Niklas Kormann[1,4]             NIKLAS.KORMANN@POLYTECHNIQUE.EDU
Masoud Ramuz[1]              MASOUD.RAMUZ@POLYTECHNIQUE.EDU
Zeeshan Nisar[2]                    NISAR@UNISTRA.FR
Nadine S. Schaadt[3]           SCHAADT.NADINE@MH-HANNOVER.DE
Benjamin Doerr[1]             BENJAMIN.DOERR@POLYTECHNIQUE.EDU
Hendrik Annuth[4]               HENDRIK.ANNUTH@FH-WEDEL.DE
Friedrich Feuerhake[3]         FEUERHAKE.FRIEDRICH@MH-HANNOVER.DE
Thomas Lampert[2]                  LAMPERT@UNISTRA.FR
Johannes F. Lutzeyer[1]        JOHANNES.LUTZEYER@POLYTECHNIQUE.EDU

[1]*LIX, École Polytechnique, IP Paris, France.* [2]*ICube, Université de Strasbourg, France.*

[3]*Institute of Pathology, Hannover Medical School, Germany.* [4]*Fachhochschule Wedel, Germany.*

**Editors:** Accepted for publication at MIDL 2025

## Abstract

Graph Neural Networks (GNNs) have recently been found to excel in histopathology. However, an important histopathological task, where GNNs have not been extensively explored, is the classification of glomeruli health as an important indicator in nephropathology. This task presents unique difficulties, particularly for the graph construction, i.e., the identification of nodes, edges, and informative features. In this work, we propose a pipeline composed of different traditional and machine learning-based computer vision techniques to identify nodes, edges, and their corresponding features to form a heterogeneous graph. We then proceed to propose a novel heterogeneous GNN architecture for glomeruli classification, called HIEGNet, that integrates both glomeruli and their surrounding immune cells. Hence, HIEGNet is able to consider the immune environment of each glomerulus in its classification. Our HIEGNet was trained and tested on a dataset of Whole Slide Images from kidney transplant patients. Experimental results demonstrate that HIEGNet outperforms several baseline models and generalises best between patients among all baseline models. Our implementation is publicly available at https://github.com/nklsKrmnn/HIEGNet.git.

**Keywords:** Graph Neural Networks, Digital Histopathology, Glomeruli Classification, Immune Environment, Graph Representation of Whole Slide Images.

## 1. Introduction

Graph Neural Networks (GNNs) have been applied in various domains to model and analyse problems involving graph-structured data. The core mechanism of GNNs, the *message passing* function, captures the relationships between entities, i.e., the topology of the data, enabling the encoding of relational information between features. In extension, heterogeneous GNNs enable learning on heterogeneous graphs, accounting for different node types. Given these capabilities, GNNs emerge as a promising alternative to Convolutional Neural Networks (CNNs) in histopathology (Brussee et al., 2025; Ahmedt-Aristizabal et al., 2021).

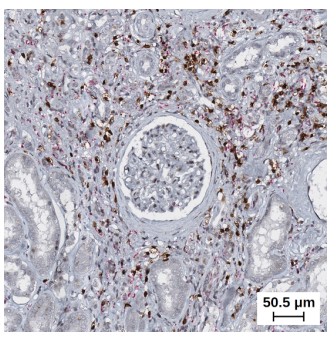
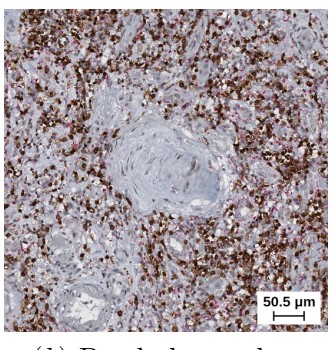
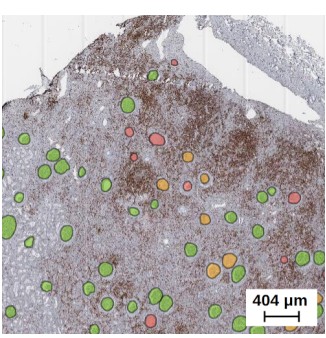

(*a*) Healthy glomerulus.      (*b*) Dead glomerulus.      (*c*) Tissue slice.

Figure 1: Images (a) and (b) show a healthy and dead glomerulus, respectively. Image (c) shows renal tissue with multiple glomeruli coloured by class membership (green: "healthy", yellow: "sclerotic", red: "dead").

In the field of medical diagnosis, histopathology is crucial to detect alterations in tissue and understand biological phenomena on the microscopic level (Welsch and Deller, 2006; Brussee et al., 2025). Glomerulosclerosis (He et al., 2024; Ayyar et al., 2018; Liu, 2006), i.e., fibrosis in glomeruli, and the often co-localized immune infiltration with macrophages and T-cells (Wynn and Barron, 2010; Xu et al., 2023; Adler et al., 2019) play an important role in chronic kidney disease (CKD), which is a major global public health concern affecting 10% of the general population worldwide (Kovesdy, 2022). The histological description and fibrotic state classification of glomeruli (Figure 1) are relevant in CKD diagnosis.

A natural approach to assist glomeruli classification is the use of CNNs. However, the application of computer vision methods to histopathology comes with challenges. The large size of Whole Slide Images (WSIs) requires patch-wise processing with CNNs (Banerji and Mitra, 2022). This limits the ability of CNNs to capture the context of the whole tissue slice. Further problems come with image domain shifts that can be introduced in many different stages of the WSI preparation process (Banerji and Mitra, 2022; Vasiljevic et al., 2021), which requires a level of generalisation that is difficult to achieve with CNNs processing images at the pixel-level. CNNs also show inefficiencies in dealing with relation-aware representations (Ahmedt-Aristizabal et al., 2021). In this work, we address these shortcomings of CNNs by proposing a graph-based approach, which explicitly models glomeruli and immune cells as nodes with associated node features. This design allows us to process entire WSIs at low computational costs and to better generalise to unseen patients. By design, our GNN is able to model cell interactions explicitly in its message passing step.

**Contributions.** Our contributions are listed as follows.

**1)** We propose a novel pipeline for constructing *heterogeneous* graphs from WSIs, incorporating glomeruli and their immune environment, represented by macrophages and T-cells. This pipeline includes image-based methods for segmenting immune cells and extracting meaningful node features. To the best of our knowledge, this is the first work that uses such an explicit representation of immune cells for glomeruli health classification.

**2)** We introduce a novel GNN architecture, called *Histopathology with Immune Environment Graph Neural Network* (HIEGNet), leveraging the heterogeneous graph construction.

**3)** Backed by experiments on real-world graphs, we show that our HIEGNet performs best among all baseline models in a setting with separated train and test patients.

## 2. Related Work

To the best of our knowledge, the first study on using deep learning for glomeruli classification was conducted by Ayyar et al. (2018), who evaluated several CNNs pre-trained on ImageNet (Krizhevsky et al., 2012), for classifying normal and abnormal glomeruli and suggested a pre-trained InceptionResNetV2 feature extractor with a Logistic Regression classifier. Altini et al. (2023) investigated CNNs to support the Oxford Classification (Cattran et al., 2009), a prominent system in nephropathology for the risk prediction of IgA nephropath, a chronic kidney disease. They tested several common CNN architectures to predict continuous scores for glomerulosclerosis and showed that EfficientNetV2 (Tan and Le, 2021) performs best, with ResNet (He et al., 2015) performing comparably well.

He et al. (2024) take a graph-based approach, including a Graph Convolutional Network (GCN) along with a Bayesian Collaborative Learning (BCL) framework for the classification of glomerular lesions. For the graph construction, He et al. obtain nodes through the creation of superpixels for each image patch, resulting in one graph per glomerulus. Node features are extracted using a pre-trained ResNet-34 (He et al., 2015) fine-tuned for histopathological images using BCL.

In contrast to previous work, we explicitly represent the immune environment in our graph construction, which we carefully developed to enable effective generalisation of our model. This allows us to propose a novel heterogeneous GNN, which, through different message passing schemes on different edge types, is able to faithfully model interaction on this heterogeneous graph to classify glomeruli.

## 3. Graph Construction

We constructed an undirected heterogeneous graph $\mathcal{G} = (\mathcal{V}, \mathcal{E}, \mathcal{T}, \mathcal{R})$ defined as a set of nodes $\mathcal{V}$ of multiple node types $\mathcal{T}$ and a set of edges $\mathcal{E}$ of different edge types $\mathcal{R}$. The node types $\mathcal{T} = \{g, m, t\}$ represent the different tissue structures with glomeruli $g$, macrophages $m$ and T-cells $t$. We now outline our proposed pipeline, including various image-based methods for obtaining the nodes, node features, and edges from the WSIs. In Appendix A, we illustrate our graph construction on an image patch containing several glomeruli.

### 3.1. Node Detection

Since image segmentation is a relevant, but separate problem from the graph construction, we assume glomeruli segmentations to be available, through manual annotation or a learning based approach, in our graph construction. We hence use each annotated glomerulus segmentation mask to define a node. Additionally, macrophages and T-cells in close proximity to glomeruli were incorporated into the graph. Previous studies on kidney WSIs (Vasiljevic et al., 2022) used immune cell nodes from squared image patches of length 554.4 µm around each glomerulus in this context. Instance segmentation was used to detect macrophages and

T-cells. To enable targeting only one of the cell types, we applied stain deconvolution to the WSIs to isolate the target staining in a distinct channel. We explored two segmentation approaches for both cell types. The first approach employs a contour detection algorithm with optimised thresholds. The second approach uses Cellpose (Stringer et al., 2020), a generalised model for instance segmentation of various cell types. In Appendix B, we describe how we optimised the threshold values for the contour detection method and fine-tuned Cellpose for both immune cell types. Furthermore, we assess their performance on a small test set of manually annotated segmentation masks. As a result, we selected Cellpose for segmenting T-cells and contour detection for segmenting macrophages.

### 3.2. Node Feature Engineering

For all nodes, we extracted node features from the WSIs. Next to the informativeness of the features regarding the classification task, we considered the following criteria to ensure reliable image feature extraction. The features should be: (1) as robust as possible to changes in the image domain, like varying colour intensities depending on illumination equipment or manufacturer lots of stainings (Banerji and Mitra, 2022); (2) invariant to rotation, as glomeruli have no orientation; (3) obtainable without manual tuning or intervention.

To satisfy these criteria, we work with Local Binary Patterns (LBPs) (Ojala et al., 1994) and shape-based features. We use uniform LBPs with a (8,1) neighbourhood to capture the homogeneity of glomeruli tissue, caused by the accumulation of extracellular matrix through fibrosis (Liu, 2006), while being robust to pixel intensity changes and rotations. To capture the deformation of glomeruli, we extracted eccentricity, circularity and aspect ratio, quantifying the roundness, in addition to the most basic shape-based measures, area and perimeter. As glomerulosclerosis does not affect the texture of immune cells, we limited their feature selection to the shape-based properties. We further added a binary feature for each immune cell that captures whether the immune cell is inside or outside of a glomerulus. All glomeruli and immune cell node features were normalised to $[0, 1]$ using min-max scaling.

### 3.3. Edge Creation

In GNNs, edges between nodes facilitate the message passing mechanism. Since there is no inherent adjacency between cells, we constructed an edge $(v, r, u) \in \mathcal{E}$ for our heterogeneous graph $\mathcal{G}$ between two nodes $v$ and $u$ depending on the Euclidean distance $d_{v,u}$ between them and on the edge type $r \in \mathcal{R}$. A specific edge type $r_{\tau_1, \tau_2}$ was defined to connect a node of type $\tau_1$ to a node of type $\tau_2$ in the given direction. The distance-based graph construction was motivated by the biological assumption that interactions between cells and cellular structures are more likely to occur if they are in close proximity within the tissue.

To select the most appropriate edge creation method for each type, we grouped the edge types into subsets, which are also visualised through different line styles in Figure 2. Edges in these subsets are all constructed following the same rules, which we describe now.

Edges between immune cell node types aim to represent the interactions between the immune cells, like the activation of macrophages by some T-cell types (Sauls et al., 2023). This subset of edge types is defined as $\mathcal{R}_i = \{r_{t,t}, r_{m,t}, r_{t,m}, r_{m,m}\} \subset \mathcal{R}$. Edges of these edge types were constructed using a $k$-nearest neighbour graph construction with $k = 5$

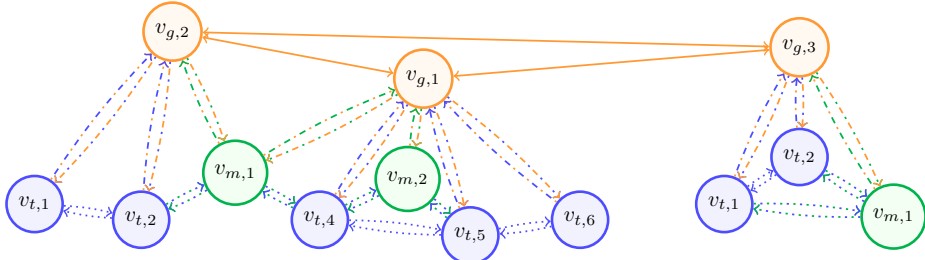

Figure 2: Illustration of a graph with three glomeruli $v_{g,i}$ (orange) and their surrounding macrophages $v_{m,j}$ (green) and T-cells $v_{t,k}$ (blue). The different line styles illustrate different message passing methods and their colour different edge types.

combined with a radius-based $\epsilon$-neighbourhood graph construction to limit edge length to a maximum of $\epsilon = 100$ µm.

Edge types connecting immune cells with glomeruli in both directions aim to exchange information between the glomeruli and their surrounding immune environment. Studies have found that the immune environment can be an indicator for the presence or absence of fibrosis in some cases (Wynn and Barron, 2010; Xu et al., 2023; Braun et al., 2021). This subset of edge types $\mathcal{R}_{ig}$ are defined as $\mathcal{R}_{ig} = \{r_{t,g}, r_{g,t}, r_{g,m}, r_{m,g}\} \subset \mathcal{R}$. We adopted findings of a previous study (Merveille et al., 2021) which identified an optimal neighbourhood radius $\epsilon = 277$ µm around glomeruli to represent the immune environment.

Edges between glomeruli $r_{g,g} \in \mathcal{R}$ are treated differently from all other edge types as well. In a small experimental set-up, which we describe in Appendix C, we found out that a $\epsilon$-neighbourhood with $\epsilon = 138.6$ µm results in the best performance with a GNN.

For all edge types, we included the Euclidean distance between the two nodes of an edge as an edge feature. The edge features were normalised using min-max scaling.

## 4. Proposed Architecture - HIEGNet

We propose HIEGNet, a novel GNN architecture for tissue health classification, which processes heterogeneous graphs such as the construction we outlined in Section 3.

GNNs are deep neural networks specifically designed to process graph-structured data. With HIEGNet, we aim to generate node representations that capture both glomeruli and immune cell features and their topological information. The *message passing* layer is the fundamental building block of every GNN, which itself consists of a *message passing function* $M(\cdot)$ and an *update function* $U(\cdot)$ (Gilmer et al., 2017). In layer $\ell$, $M(\cdot)$ computes a message $\boldsymbol{m}_v^{(\ell)}$ for each node $v \in \mathcal{V}$ from the hidden embeddings $\boldsymbol{h}^{(\ell-1)}$ of the node $v$ itself and its neighbours $u \in \mathcal{N}(v)$ of the previous layer.

$$\boldsymbol{m}_v^{(\ell)} = M\big(\boldsymbol{h}_v^{(\ell)}, \{\boldsymbol{h}_u^{(\ell)} : u \in \mathcal{N}(v)\}\big),$$

where $\mathcal{N}(v) = \{u \in \mathcal{V} : (v, r, u) \in \mathcal{E}\}$. These embeddings are aggregated for each node via a permutation-invariant aggregation function. The update function $U(\cdot)$ uses the message

$\boldsymbol{m}_v^{(\ell)}$ and the hidden state $\boldsymbol{h}_v^{(\ell)}$ to compute the next hidden state $\boldsymbol{h}_v^{(\ell+1)}$

$$\boldsymbol{h}_v^{(\ell+1)} = U\big(\boldsymbol{h}_v^{(\ell)}, \boldsymbol{m}_v^{(\ell)}\big).$$

The function $U(\cdot)$ is an arbitrary trainable function, which we implemented as multi-layer perceptrons (MLPs). For node classification, we passed the final node representations $\boldsymbol{h}_v^{(L)}$ through a fully connected layer with a Softmax activation function afterwards, which is trained end-to-end with the GNN. This node-wise message passing and classification head preserves the invariance to Euclidean symmetries, such as rotation of the WSI, and by definition, enables the explicit representation of glomeruli and immune cells introduced by our graph-based representation of the WSIs.

The message passing mechanism is the central design element that distinguishes different GNNs. In this work, we employed different message passing functions, selected to align with the characteristics of the different edge types grouped as in Section 3.3: $\{r_{g,g}\}$, $\mathcal{R}_i$, and $\mathcal{R}_{ig}$. Within each edge type group, a single message passing function was applied. However, the learnable parameters were shared only among edges of the same specific type $r \in \mathcal{R}$. These edge-specific parameter sets allow for a simple implementation of additional cell types, such as B-cells, by simply adding them as node types with corresponding edge types. For the aggregation of messages from different edge types, we extended the summation mechanism used in the Relational Graph Convolutional Network (RGCN) (Schlichtkrull et al., 2018) to incorporate different message passing functions. Hence, we define a message passing layer of HIEGNet as

$$\boldsymbol{h}_v^{(\ell+1)} = \sum_{r\in\mathcal{R}} U_r^{(\ell)}\Big(\boldsymbol{h}_v^{(\ell)}, M_r^{(\ell)}\Big(\boldsymbol{h}_v^{(\ell)}, \{\boldsymbol{h}_u^{(\ell)} : (v,r,u)\in\mathcal{E}\}\Big)\Big), \tag{1}$$

where $U_r^{(\ell)}(\cdot)$ and $M_r^{(\ell)}(\cdot)$ are update and message functions specific to the edge type $r$ with a separate parameter set for each layer $\ell$. In Figure 2, the different model parameter sets are illustrated through different colour combinations on the edges. To exemplify our HIEGNet, we now provide the model equation of an instance of our HIEGNet, where different message passing schemes are used for the defined edge type subsets.

$$\begin{aligned}
\boldsymbol{h}_v^{(\ell+1)} =& \sigma\left(\boldsymbol{W}_{r_{g,g}}^{(\ell)} \sum_{u\in\mathcal{N}(v)\cup\{u\}} \alpha_{v,u,r}\boldsymbol{h}_u^{(\ell)}\right) \\
&+ \sum_{r\in\mathcal{R}_{ig}} \sigma\left(\boldsymbol{W}_{U,r}^{(\ell)}\sigma\left(\boldsymbol{W}_{M,r}^{(\ell)}\left[\boldsymbol{h}_v^{(\ell)}|| \sum_{u\in\mathcal{N}(v)} \frac{\boldsymbol{h}_v^{(\ell)}}{|\mathcal{N}(v)|}\right]\right)\right) \\
&+ \sum_{r\in\mathcal{R}_i} \sigma\left(\boldsymbol{W}_{U,r}^{(\ell)}\sigma\left(\boldsymbol{W}_{M,r}^{(\ell)}\left[\boldsymbol{h}_v^{(\ell)}|| \sum_{u\in\mathcal{N}(v)} \frac{\boldsymbol{h}_v^{(\ell)}}{|\mathcal{N}(v)|}\right]\right)\right),
\end{aligned}$$

where $||$ denotes concatenation, $\sigma$ is an activation function, $\boldsymbol{W}_{r_{g,g}}^{(\ell)}$ and $\boldsymbol{W}_{U,r}^{(\ell)}$ are weight matrices, $\boldsymbol{W}_{M,r}^{(\ell)}$ are the weights for the GraphSAGE message passing step (Hamilton et al., 2017), and $\alpha$ is the Graph Attention model's attention coefficient (Brody et al., 2022).

We also explored the usage of Jumping-Knowledge and a modified U-Net pre-trained on WSIs (Nisar and Lampert, 2024) for feature extraction. These model variations are illustrated in Figure 3 and are further described and evaluated in Appendix D.

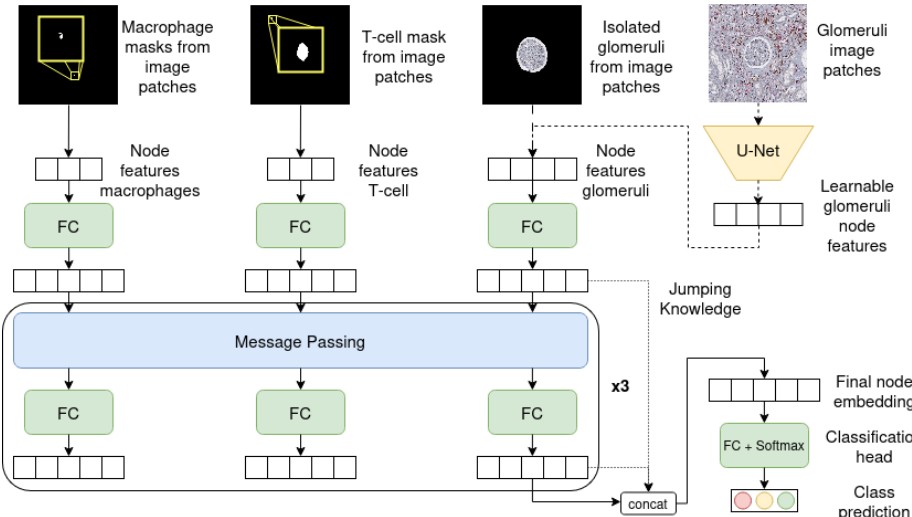

Figure 3: An illustration of all model variations. The regular HIEGNet uses only hand-crafted node features. In a hybrid model, the CNN output replaces the glomeruli node features. *Jumping knowledge* concatenates node embeddings from all layers.

## 5. Experiments

We now first describe our experimental set-up before reporting and discussing the results.

**Data.** We test our HIEGNet on histopathological data of six real patients using the EXC dataset (Merveille et al., 2021). Despite this small number of patients, it provides a sufficient number of samples, with a total of 2052 glomeruli, which is considerable given the amount of manual labour required for annotation. Each glomerulus was labelled with the class "healthy" if no evidence of fibrosis is present; "sclerotic" if it shows signs of fibrosis, but functional tissue is still present; and "dead" if no functional tissue remains. We used two experimental settings: (1) in the *within patients* setting, glomeruli from patients 001–003 were split into a training and a test set with 85% and 15% of the glomeruli, respectively. (2) in the *between patients* setting, the model is trained on the full graphs constructed from patients 001–003 and tested on the graphs from patients 004-006. This setting evaluates the model's ability to generalise to unseen patients. We optimise all models, including a hyperparameter search, for setting (1) and evaluate the top-performing models from setting (1) in both settings (1) and (2). Appendix E provides more detail on the dataset, the exact numbers of train and test sets for both settings, an analysis of the class distribution.

**Baselines.** In this study, we compared our results with several computer vision models as baseline methods, alongside a Random Forest utilising the extracted glomerular features described in Section 3.2. As suggested by Altini et al. (2023), we used a pretrained ResNet and EfficientNetV2 as baseline models. Additionally, we compared against the modified pre-trained U-Net. For further details, we refer to Appendix F.

**Evaluation Metrics.** We optimised the models for the best macro-averaged F1-Score, to account for both the class imbalance and the fact that for pathologists, detecting poten-

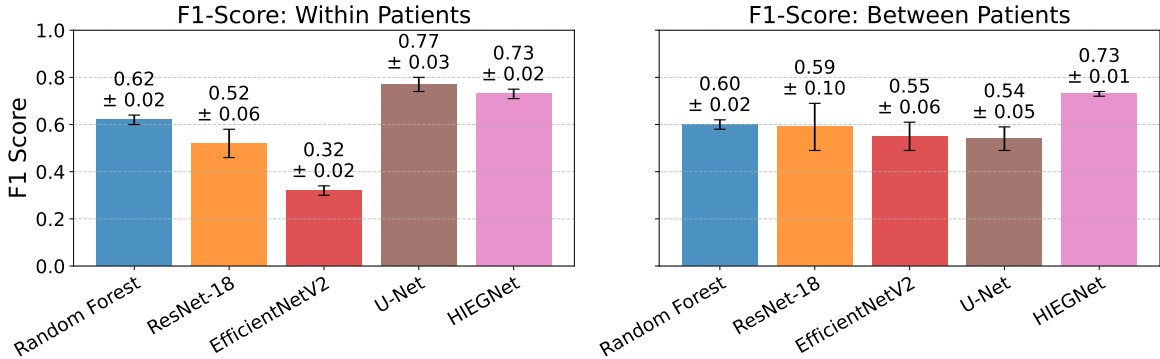

Figure 4: F1-Scores macro-averaged of all models evaluated on the test sets.

tially dead glomeruli is as important as the precision of the predictions. The underlying recall and precision scores are reported in Appendix G. The mean and standard deviation of the scores were obtained with 20 random parameter initialisations.

**Implementation.** We determined optimal hyperparameters via grid search with 4-fold cross-validation. The search space for the grid search and the optimal hyperparameters are outlined in Table 7 in Appendix H. Even though the graphs are quite large, with up to 4 million edges, the precise feature selection allows for efficient training with only 1.21 s per epoch. Further details about the computational complexity are reported in Appendix I. Our implementation is publicly available on GitHub.

**Results and Discussion.** Figure 4 presents the results for the proposed GNN and the baseline models on the EXC dataset. In the *within patients* setting, the pre-trained U-Net achieves the highest F1-Score of 0.77, slightly outperforming HIEGNet, which attains an F1-Score of 0.73. The other baselines exhibit significantly lower F1-Scores, with EfficientNet and ResNet below 0.55 and the Random Forest at 0.62. For the *between patients* setting, the HIEGNet has the overall best performance with an F1-Score of 0.73. The Random Forest demonstrates the smallest performance gap between the two settings, showing that the features we define enable good generalisation. While the performance of both the GNN, ResNet-18 and the U-Net decreases significantly when tested on unseen patients, we observe the performance drop of the GNN to be much smaller than for the other two models. The overall low performance of EfficientNet and ResNet and the strong performance decrease of the U-Net on the between patients setting indicate the effectiveness of domain-specific pre-training. The strong performance of HIEGNet compared to the Random Forest shows that the integration of a graph structure adds value for glomeruli health classification. Additionally, the representations learned by the GNN generalise better to unseen patients than image-based representations.

## 6. Conclusions

We have introduced HIEGNet, a novel GNN architecture for tissue health classification, used here for glomeruli health, by leveraging a heterogeneous graph representation of glomeruli and their surrounding immune cells as nodes. To construct this graph from WSIs, we pro-

posed a pipeline encompassing image-based segmentation and feature extraction as well as a task-specific edge construction. Experimental results demonstrate that HIEGNet surpasses established CNN baseline models in the ability to generalise across patients, highlighting the potential of GNNs as a suitable alternative to CNNs, deserving of further study and optimisation for this task.

**Future Work.** We see further potential in the development of our approach to become more robust to changes in the image domain. As discussed in Appendix J, pre-training with varying staining protocols improves the ability to generalise to image domain shifts for the baseline U-Net and is also a promising future direction of development for HIEGNet. Furthermore, to aid in the adoption of our approach, we also recommend future work on explainability methods.

## Acknowledgments

We thank Odyssée Merveille for insightful discussions towards the beginning of the project. The work of Niklas Kormann, Zeeshan Nisar, Johannes Lutzeyer and Thomas Lampert was supported by the ANR PRC HistoGraph grant (ANR-23-CE45-0038). Masoud Ramuz was supported by the Institut Carnot Télécom & Société numérique. Zeeshan Nisar, Friedrich Feuerhake, Nadine S. Schaadt and Thomas Lampert furthermore acknowledge funding by the ERACoSysMed & e:Med initiatives by BMBF, SysMIFTA (managed by PTJ, FKZ 031L-0085A; ANR, grant ANR-15-CMED-0004) and SYSIMIT (managed by DLR, FKZ01ZX1308A) for the high-quality images and thank Nvidia, the *Centre de Calcul* (University of Strasbourg) & GENCI-IDRIS (grant 2020-A0091011872) for GPU access.

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

# Appendix A. Visualisation of the Graph Construction

Figure 5 illustrates the graph construction with the pipeline proposed in Section 3.

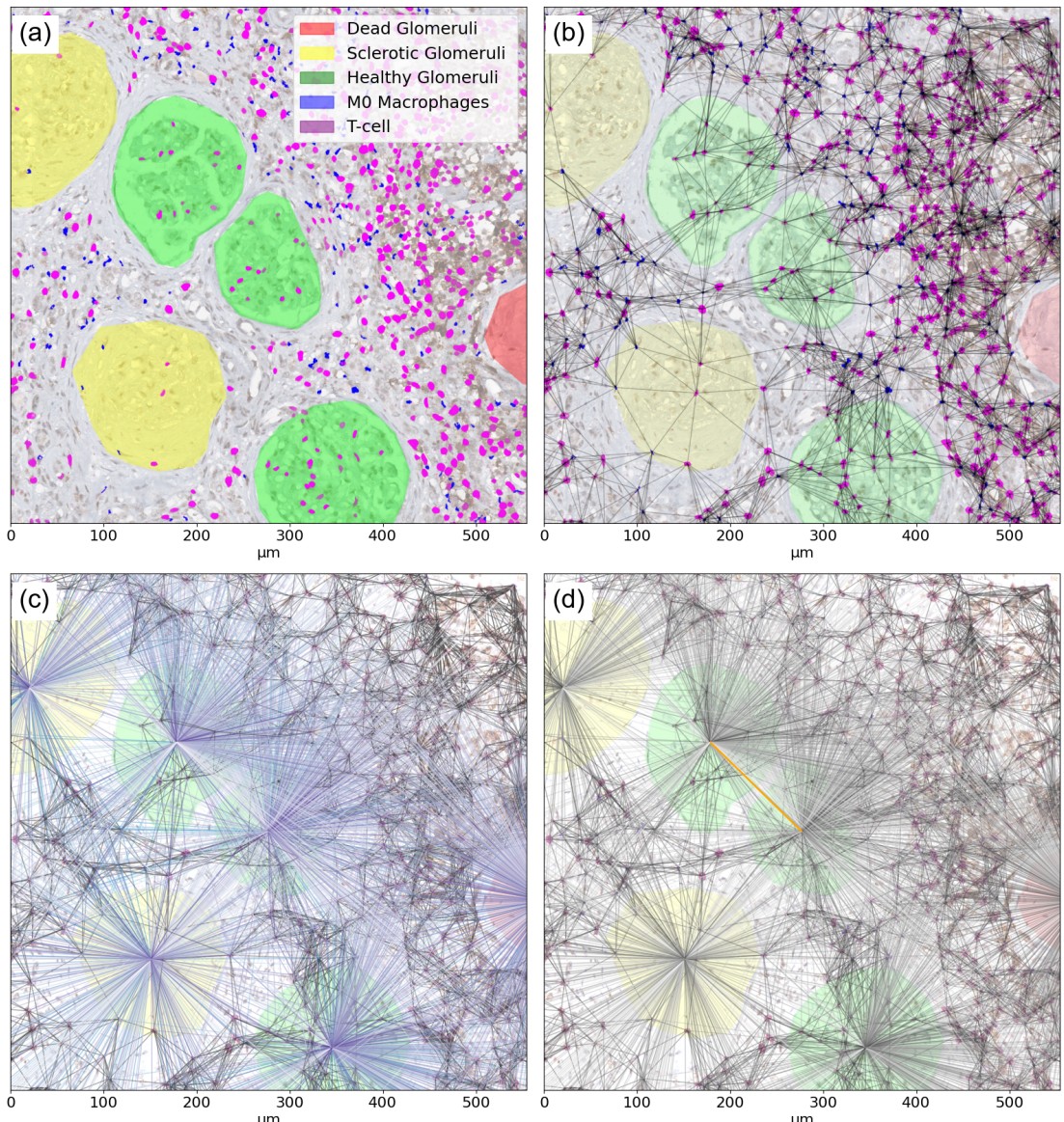

Figure 5: Step-by-step graph construction from a WSI patch of a given glomerulus. (a) WSI with glomeruli, T-cells and macrophage mask. (b) Overlay of edges of type $r \in \mathcal{R}_i$. (c) Overlay of edges of type $r \in \mathcal{R}_{ig}$. Edges of type $r_{t,g}$ and are coloured in blue and purple, respectively. The colour saturation decreases as the length of the edge increases, making longer *edges* appear closer to white. (d) Overlay of edges of type $r_{g,g}$ in orange.

## Appendix B. Evaluation of the Cell Segmentation Method

For evaluating the cell segmentation methods, we manually annotated segmentation masks of all T-cells and macrophages in a subset of 12 squared WSI patches of length 554.5 µm around a glomerulus. This subset included one patch per glomerulus class for each patient to make is as representative as possible while keeping the annotation workload manageable. Details of the selected image patches, including the number of annotation masks per image and their allocation to the training and test sets for both experimental settings, are summarised in Table 1.

Table 1: Number of cell annotations per immune cell type for each image contained in the subset for training and evaluation of the cell segmentation methods.

| ID | Patient | Class | Number of Masks | | Set Assignment | |
|---|---|---|---|---|---|---|
| | | | M0 | T-cell | Within | Between |
| 1330965 | 001 | Healthy | 270 | 305 | Train | Train |
| 1332018 | 001 | Sclerotic | 235 | 926 | Train | Train |
| 1333320 | 001 | Sclerotic | 345 | 464 | Test | Train |
| 1334171 | 001 | Dead | 143 | 302 | Train | Train |
| 1026227 | 002 | Healthy | 156 | 364 | Test | Train |
| 1067488 | 002 | Dead | 191 | 612 | Train | Train |
| 994858 | 003 | Healthy | 140 | 232 | Train | Train |
| 995327 | 003 | Sclerotic | 135 | 227 | Train | Train |
| 995518 | 003 | Dead | 156 | 280 | Test | Train |
| 984846 | 004 | Healthy | 120 | 497 | - | Test |
| 988740 | 004 | Sclerotic | 95 | 644 | - | Test |
| 985159 | 004 | Dead | 106 | 512 | - | Test |

We evaluated three methods for segmenting immune cells in the tissue scans:

1. Fine-tuning Cellpose on the training set, using a learning rate of 0.001, a weight decay of 0.01, and training for 100 epochs, following the recommendations of the authors (Stringer et al., 2020).

2. Using a pre-trained, frozen Cellpose model without any fine-tuning on the EXC dataset.

3. Applying a contour detection algorithm (Suzuki and Abe, 1985), which used two thresholds: an intensity threshold to filter out weak staining signals and an area threshold to exclude contours too small to represent target T-cells.

For the deep learning-based segmentation method, we trained two separate pre-trained Cellpose 2.0 (Stringer and Pachitariu, 2022) models, one for macrophages and one for T-cells. These models were trained using the annotated masks from the training sets in both the within patient and between patient settings. In contrast, the contour detection method did not require model training. Instead, its thresholds were optimised on the training

set. Specifically, the intensity threshold was tuned with values ranging from 10 to 240 in increments of 10, based on a maximum intensity of 255. The area threshold was optimised with values ranging from 40 to 200 pixels in increments of 20, corresponding to areas of 2.5 to 12.7 µm². The optimal threshold values were selected based on the area under the curve (AUC) of average precision (AP) scores evaluated on the training set. For the AUC the AP scores were computed for intersection over union (IOU) thresholds ranging from 0.05 to 1.0 in increments of 0.05.

We compared the different segmentation methods based on the AUC thresholds for T-cells and macrophages across both experimental settings. The results are presented in Table 2 and Table 3, respectively. For T-cell segmentation, the results show that the fine-tuned Cellpose model outperforms the contour detection method and the frozen model in all cases. However, for macrophage segmentation, the contour detection method outperformed both the frozen and fine-tuned Cellpose models in most scenarios. The final thresholds were determined on the test set within patients 001 - 003 with an intensity threshold value of 60 and an area threshold of 160 pixels. Further investigations revealed that setting the flow threshold, which validates predictions based on gradient flow (Stringer and Pachitariu, 2020), to 0.0 was necessary to generate any predictions for macrophages, underscoring the limitations of the fine-tuned Cellpose model in this context.

Table 2: AUC values scored with the three evaluated segmentation methods for T-cells on the test set for the within patients setting and four between patients settings using each of the four patients as test patient.

| | Within patients | Between patients | | | |
|---|---|---|---|---|---|
| Test patients | 001-003 | 001 | 002 | 003 | 004 |
| Contour detection | 0.233 | 0.324 | 0.342 | 0.421 | 0.233 |
| Frozen Cellpose | 0.267 | 0.219 | 0.308 | 0.225 | 0.267 |
| Fine-tuned Cellpose | **0.525** | **0.493** | **0.548** | **0.519** | **0.525** |

Table 3: AUC values scored with the three evaluated segmentation methods for M0 macrophages on the test set for the within patients setting and four between patients settings using each of the four patients as test patient.

| | Within patients | Between patients | | | |
|---|---|---|---|---|---|
| Test patients | 001-003 | 001 | 002 | 003 | 004 |
| Contour detection | **0.265** | **0.003** | **0.003** | 0.134 | **0.202** |
| Frozen Cellpose | 0.190 | 0.000 | 0.002 | **0.268** | 0.120 |
| Fine-tuned Cellpose | 0.000 | 0.000 | 0.000 | 0.000 | 0.000 |

## Appendix C. Graph Construction between Glomeruli

To find a suitable graph construction method for edges between glomeruli of type $r_{g,g}$ we tested $\epsilon$-neighbourhood graph construction and $k$-NN graph construction with different value for $\epsilon$ and $k$ for different message passing methods $M_{r_{g,g}}(\cdot)$ and different dropout rates $p$. Table 4 contains mean macro-averaged F1-Scores from a 4-fold cross validation on the training set for all these combinations of $M_{r_{g,g}}(\cdot)$, $p$ and the graph construction methods. A graph with edges of type $r_{g,g}$ constructed with a radius-based method with $\epsilon = 550$ pixels $= 227$ μm, achieves the highest mean score of all hyperparameter combination and outperforms all other methods for 75% of all cases. Therefore, we used this construction setting for our proposed pipeline.

Table 4: Macro-averaged F1-Scores from a 4-fold cross validation on the training set for different graph construction methods for edges of type $r_{g,g}$ with various hyperparameter combinations. The values for the radius $\epsilon$ are given in pixels.

| $M_{r_{g,g}}(\cdot)$ | $p$ | $k=1$ | $k=2$ | $k=3$ | $k=4$ | $k=5$ | $\epsilon=3000$ | $\epsilon=5000$ | $\epsilon=550$ |
|---|---|---|---|---|---|---|---|---|---|
| | | **$k$NN-Graph** | | | | | **$\epsilon$-Neighbourhood Graph** | | |
| GATv2 | 0.00 | **0.43** | 0.39 | 0.39 | 0.42 | 0.42 | 0.37 | 0.33 | 0.31 |
| GATv2 | 0.05 | 0.58 | 0.54 | 0.54 | 0.51 | 0.50 | 0.54 | 0.51 | **0.75** |
| GATv2 | 0.10 | 0.59 | 0.54 | 0.52 | 0.54 | 0.53 | 0.53 | 0.55 | **0.75** |
| GATv2 | 0.15 | 0.63 | 0.58 | 0.58 | 0.53 | 0.56 | 0.55 | 0.54 | **0.74** |
| GCN | 0.00 | **0.50** | 0.46 | 0.49 | 0.49 | 0.49 | 0.47 | 0.48 | 0.19 |
| GCN | 0.05 | 0.56 | 0.53 | 0.56 | 0.53 | 0.49 | 0.51 | 0.54 | **0.69** |
| GCN | 0.10 | 0.61 | 0.60 | 0.58 | 0.53 | 0.49 | 0.53 | 0.52 | **0.71** |
| GCN | 0.15 | 0.65 | 0.56 | 0.56 | 0.52 | 0.50 | 0.53 | 0.59 | **0.72** |
| GIN | 0.00 | 0.42 | **0.51** | 0.43 | 0.48 | 0.48 | 0.45 | 0.44 | 0.25 |
| GIN | 0.05 | 0.53 | 0.56 | 0.52 | 0.50 | 0.48 | 0.49 | 0.50 | **0.74** |
| GIN | 0.10 | 0.53 | 0.57 | 0.51 | 0.52 | 0.51 | 0.49 | 0.48 | **0.72** |
| GIN | 0.15 | 0.58 | 0.55 | 0.50 | 0.52 | 0.50 | 0.50 | 0.48 | **0.70** |
| **Mean** | | 0.55 | 0.53 | 0.52 | 0.51 | 0.50 | 0.50 | 0.50 | **0.61** |

## Appendix D. Model Variations

In addition to the HIEGNet architecture proposed in Section 4 we developed and tested further variations of the model, which are also illustrated in Figure 3, incorporating *jumping knowledge* (Xu et al., 2018) as a HIEGNet-JK and learned features as a Hybrid-HIEGNet.

In HIEGNet-JK, all hidden embeddings of a node, including the initial node embedding, are concatenated as the final node embedding before applying the FC layer for classification. This model variation was optimised with the same grid search space presented in Appendix H for HIEGNet. As a result of the grid search, we identified a dropout rate of $p = 0.2$,

hidden dimension of $d_h = 64$, number of message passings or layers $L = 3$ and number of fully connected layers in $U(\cdot)$ of 1 as optimal hyperparameters for HIEGNet-JK. Further, we identified E-SAGE for $r_{g,g}$, GATv2 for $r \in \mathcal{R}_{ig}$, and GINE for $r \in \mathcal{R}_i$ to be the optimal message passing function.

The Hybrid-HIEGNet employs a CNN for feature extraction instead of engineered glomeruli features. Such a hybrid model is more general since it can be adopted for other tasks that involve similar consideration of the immune environment but with different objects to classify, without domain-specific knowledge required. However, this hybrid model comes with the challenges of CNNs in histopathology we discussed in Section 1.

The hybrid model developed in this work generates node features from each glomerulus using a modified pre-trained U-Net (Ronneberger et al., 2015) applied to the corresponding image patch, as illustrated in Figure 3. Nisar and Lampert (Nisar and Lampert, 2024) explored various self-supervised learning techniques to optimise a U-Net architecture for glomeruli segmentation in WSIs with diverse staining protocols. For this work, a U-Net pre-trained using BYOL (Grill et al., 2020) on a dataset of histopathological images, which is described in (Nisar and Lampert, 2024) is used as a feature extractor. It was fine-tuned on our dataset and modified for classification by applying Global Average Pooling and a 3-layer MLP to the final hidden feature map.

This model variation was optimised with the same grid search space presented in Appendix H for HIEGNet with one adjustment. For the hidden dimension we defined the search space as {64, 128}. As a result of the grid search, we identified a dropout rate of $p = 0.4$, hidden dimension of $d_h = 128$, number of message passings or layers $L = 3$ and number of fully connected layers in $U(\cdot)$ of 1 as optimal hyperparameters for Hybrid-HIEGNet. Further, we identified SAGE for $r_{g,g}$, GATv2 for $r \in \mathcal{R}_{ig}$, and SAGE for $r \in \mathcal{R}_i$ to be the optimal message passing function.

The results for both models are reported in Figure 6.

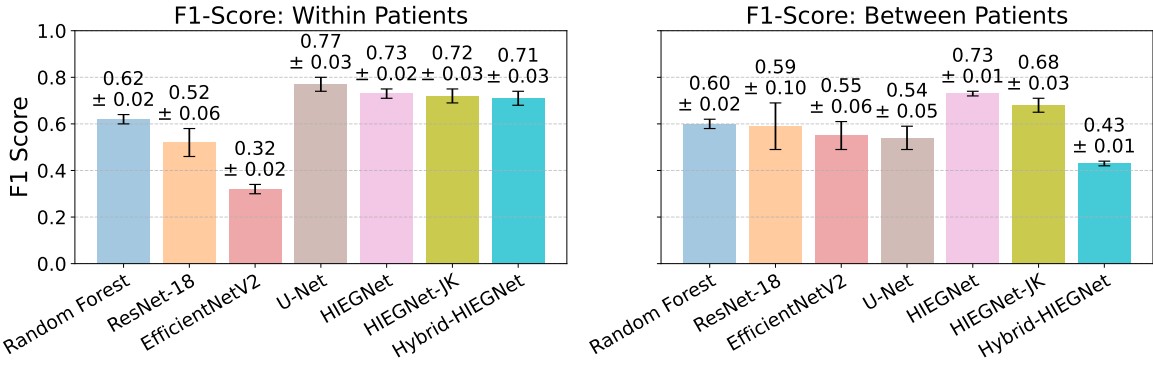

Figure 6: F1-Scores macro-averaged of all models including HIEGNet-JK and Hybrid-HIEGNet evaluated on the test sets.

## Appendix E. Dataset Overview

The EXC dataset comprises kidney tissue samples extracted through excision biopsies from six patients who experienced rejection of a kidney transplant. As a result of the rejection, all four patients are affected by glomerulosclerosis and show signs of "hot fibrosis", which some studies use to refer to the state of fibrosis with immune cell activity (Adler et al., 2019). The slices are stained with a combination of haematoxylin, 3,3'-Diaminobenzidine (DAB) for CD3 (brown), and new fuchsine for CD68 (red). Haematoxylin is a histochemical counterstain that highlights cell nuclei and other general tissue structures. CD3 and CD68 are specialised markers highlighting T-cells and M0 macrophages, respectively. The WSIs were acquired using a slide scanner AT2 from Leica Biosystems with a 40x magnification, resulting in a resolution of 0.252 µm/pixel. For further details on the dataset, we refer to Merveille et al. (2021).

Annotations of the glomeruli were obtained through a consensus-based manual labelling approach. After initial annotations based on strict characteristics for the three classes:

- Healthy: Intact Bowman capsule, clearly visible blood vessels throughout the glomerulus, presence of immune cells, and no extracellular matrix.

- Dead: Absence of Bowman capsule, over $50\%$ of the glomerulus filled with extracellular matrix, minimal or no visible blood vessels, and no immune cells.

- Sclerotic: Damaged or absent Bowman capsule, partially visible blood vessels, and extracellular matrix involvement in less than 50

We discussed edge cases where characteristics of different classes can be found in one glomerulus (heavily relying on the pathologists in our consortium) until we came to a consensus. To further safeguard label quality, we excluded glomeruli with profiles smaller than 30µm in diameter. This step prevented the inclusion of cases where partial sectioning might lead to incomplete or misleading morphological assessments.

The dataset is unbalanced with 1326 glomeruli labelled as "healthy" and only 417 and 307 glomeruli labelled as "sclerotic" and "dead", respectively. Furthermore, the class distribution varies significantly between the six patients, as visualised in Figure 7A. Therefore, we used a stratified test split for evaluation and stratified cross validation for patients 001-003 in the between patients setting. Table 5 provides the exact numbers of train and test sets for both settings and Figure 7B visualises the class distribution for both settings.

## Appendix F. Further Details on the Baseline Models

We optimised all baseline models to the best of our abilities through a combination of initial experiments and grid search. The search space as well as the final hyperparameter setting are:

- **ResNet.** We used the ResNet pre-trained on ImageNet available in PyTorch, initialised a classification head of 3 FC layers and fine-tuned it for 300 epochs using a batch size of 20 images, due to memory capacity limits. We used a weighted cross entropy loss function to address class imbalance and used a One-Cycle learning rate scheduler with an initial learning rate of 2.5e-5. We tested dropout rates

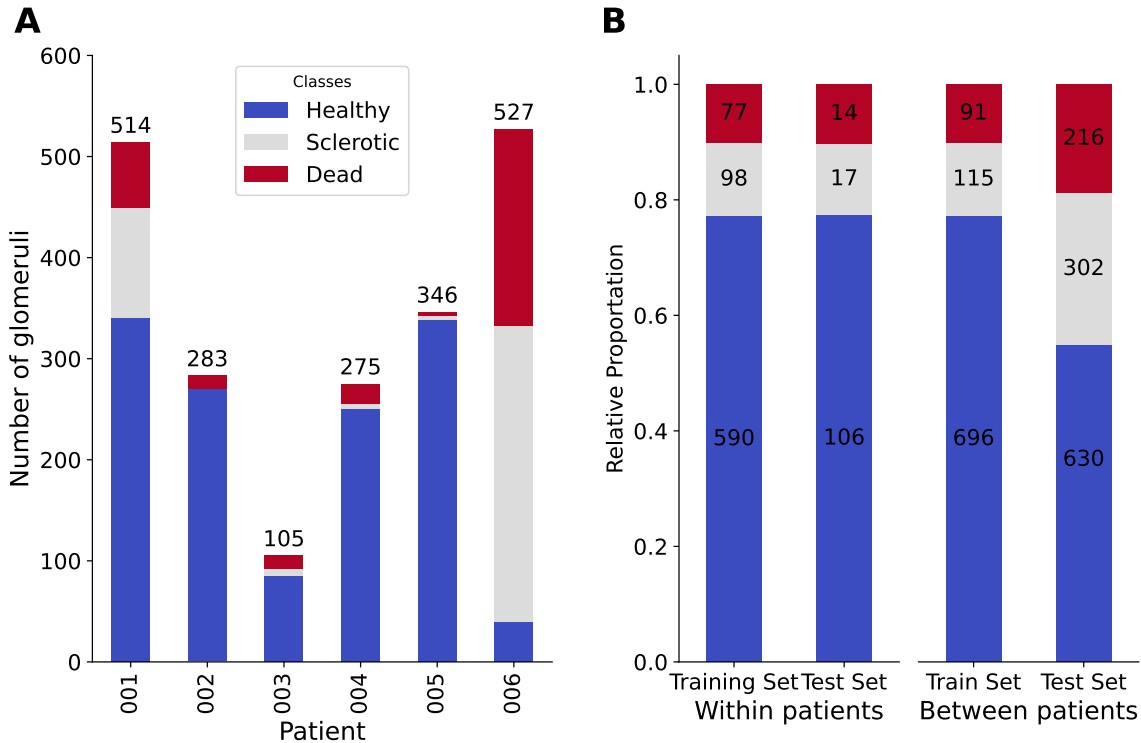

Figure 7: **A** Absolute distribution of samples over classes for each patient. **B** Relative distribution of samples over classes in the train and the test set for within patients and between patients settings, with absolute quantities as bar annotation.

Table 5: Overview of train and test set for within patients and between patients settings.

| Patient | Within patients | | Between patients | |
|---|---|---|---|---|
| | **Train set** | **Test set** | **Train set** | **Test set** |
| 001 | 447 (85%) | 78 (15%) | 515 (100%) | - (0%) |
| 002 | 240 (85%) | 43 (15%) | 283 (100%) | - (0%) |
| 003 | 89 (85%) | 16 (15%) | 105 (100%) | - (0%) |
| 004 | - (0%) | - (0%) | 0 (0%) | 275 (100%) |
| 005 | - (0%) | - (0%) | 0 (0%) | 346 (100%) |
| 006 | - (0%) | - (0%) | 0 (0%) | 528 (100%) |
| Sum | 766 (85%) | 137 (15%) | 903 (44%) | 1,149(56%) |

$p \in \{0.2, 0.4, 0.6, 0.7, 0.8\}$ and number of layers $L \in \{18, 34\}$. The best mean F1-Score was achieved using $p = 0.4$ and $L = 18$.

- **EfficientNetV2.** We used the EfficientNetV2 pre-trained on ImageNet available in PyTorch, initialised a classification head of 3 FC layers and fine-tuned it for 300 epochs using a batch size of 14 images, due to memory capacity limits. We used a weighted loss cross entropy function to address class imbalance and used a One-Cycle learning rate scheduler with an initial learning rate of 2.5e-5. We tested dropout rates $p \in \{0.2, 0.4, 0.6, 0.8\}$ and the model size $s \in \{$"s", "m"$\}$. The best mean F1-Score was achieved using $p = 0.2$ and $s = $ "m".

- **U-Net.** We used the U-Net from (Nisar and Lampert, 2024) pre-trained on a dataset of histopathological images, applied global average pooling to the final feature maps, initialised a classification head of 3 FC layers and fine-tuned it for 200 epochs with a dropout rate of $p = 0.4$ and a batch size of 20 images, due to memory capacity limits.

- **Random Forest.** We used the Random Forest implemented in Sci-kit learn with all node features used for the glomeruli nodes as described in Section 3.2. We tested the number of estimators $n_{trees} \in \{100, 500, 1000\}$ and maximal number of features $n_{ft\_max} \in \{4, 15\}$, maximal depth $d_{max} \in \{10, 50, 100, \inf\}$, minimal number of samples for a split $n_{min\_split} \in \{2, 5, 10\}$ and the minimal number of samples per leaf $n_{min\_leaf} \in \{1, 2, 4\}$. The best mean F1-Score was achieved using $n_{trees} = 100$, $n_{ft\_max} = 4$, $d_{max} = 10$, $n_{min\_split} = 2$, and $n_{min\_leaf} = 1$.

All three image-based baseline models were fine-tuned using standardised bright-field glomerular image patches from our dataset as input. For the training, we used the ADAM optimiser, a (maximal) learning rate of 0.0001 and the cross-entropy loss function. All hyperparameter sets were optimised in a grid search with 4-fold cross-validation to ensure robust validation results.

## Appendix G. Additional Score Results

To complement the reported F1-Scores in Section 5 we report the macro-average precision and recall for all models in Table 6.

Table 6: Precision and recall scores for the experiments without augmentations.

| Model | Within patients | | Between patients | |
|---|---|---|---|---|
| | Precision | Recall | Precision | Recall |
| Random Forrest | $0.66 \pm 0.02$ | $0.63 \pm 0.02$ | $0.70 \pm 0.01$ | $0.62 \pm 0.01$ |
| ResNet-18 | $0.49 \pm 0.06$ | $0.71 \pm 0.07$ | $0.- \pm 0.-$ | $0.- \pm 0.-$ |
| EfficientNetV2 | $0.33 \pm 0.02$ | $0.33 \pm 0.05$ | $0.- \pm 0.-$ | $0.- \pm 0.-$ |
| U-Net | $\mathbf{0.88 \pm 0.03}$ | $\mathbf{0.76 \pm 0.04}$ | $0.66 \pm 0.03$ | $0.51 \pm 0.06$ |
| HIEGNet | $0.78 \pm 0.01$ | $0.65 \pm 0.03$ | $\mathbf{0.73 \pm 0.01}$ | $\mathbf{0.74 \pm 0.01}$ |
| HIEGNet-JK | $0.76 \pm 0.04$ | $0.68 \pm 0.02$ | $0.70 \pm 0.03$ | $0.70 \pm 0.01$ |
| Hyrbid-HIEGNet | $0.70 \pm 0.03$ | $0.65 \pm 0.02$ | $0.44 \pm 0.01$ | $0.54 \pm 0.02$ |

## Appendix H. Gird Search Space

Next to commonly optimised hyperparameters of GNNs, we aimed to identify the optimal message passing functions for the different edge type groups defined in Section 3.3. Table 7 describes the search space we covered with a grid search in the *within patients* setting, using the following abbreviations for common message passing methods: Graph Convolutional Network (GCN) (Kipf and Welling, 2017), Graph Isomorphism Network with edge features (GINE) (Hu et al., 2020), continuous-filter convolution (CFconv) from SchNet (Schütt et al., 2017), Attention Network V2 (GATv2) (Brody et al., 2022), and the message passing function of GraphSAGE (SAGE) (Hamilton et al., 2017). Each model was trained for up to 600 epochs, with early stopping employed, using ADAM optimiser and a One-Cycle learning rate scheduler with a maximal learning rate of 0.001. Each batch comprised all nodes of one graph.

Table 7: Hyperparameter search space for HIEGNet.

| Hyperparameter | Search space |
|:---:|:---:|
| Dropout rate | {0.1, 0.2, 0.3, 0.4, 0.5} |
| Hidden dimension | {32, 64} |
| Number of message passings | {2, 3} |
| Number of FC layer in $U(\cdot)$ | {1, 2} |
| Message passing methods for $r_{(r,r)}$ | {GATv2, SAGE} |
| Message passing methods for $\mathcal{R}_{ig}$ | {GATv2, SAGE} |
| Message passing methods for $\mathcal{R}_i$ | {GATv2, SAGE, GCN, GINE, CFconv} |

With this grid search we identified an optimal hyperparameter setting for for HIEGNet on this dataset. Especially, we identified the following message passing functions for each group of edge types $r \in \mathcal{R}$ in order to exploit the immune environment's graph representation for the classification of glomeruli health:

- For edges between glomeruli of type $r_{g,g}$ we use the enhanced version of GraphSAGE (Hamilton et al., 2017) message passing mechanism, which applies additional learnable weights to the hidden states of neighbouring nodes before aggregation. We also included the update function of GraphSAGE, which applies a fully connected (FC) layer to the message concatenated with the hidden state of the central node.

- For edges between glomeruli and immune cells of the types $r \in \mathcal{R}_{ig}$ we apply the GATv2 (Brody et al., 2022) message passing mechanism, which computes an attention score based on the features of the central node $\boldsymbol{h}_v^{(\ell)}$, the neighbour node $\boldsymbol{h}_u^{(\ell)}$ and the edge $\boldsymbol{e}(v,u)^{(\ell)}$ to perform a weighted sum of the neighbouring hidden states.

- For edges between immune cells of types $r \in \mathcal{R}_i$, we use the CFconv message passing mechanism introduced with SchNet (Schütt et al., 2017), which simply sums the hidden states of neighbouring nodes weighted by the inverse of the edge distance.

Further, we identified a dropout rate of $p = 0.2$, hidden dimension of $d_{\boldsymbol{h}} = 64$, number of message passings or layers $L = 2$ and number of fully connected layer in $U(\cdot)$ of 2 as optimal hyperparameters.

## Appendix I. Computational Costs and Scalability

The computational complexity of Graph Neural Networks (GNNs) is linear in the number of edges in the graph. This computational efficiency is particularly important for whole-slide images, where a single image may contain hundreds of thousands of immune cells and hundreds of glomeruli. In our pipeline, we mitigate computational overhead by limiting the node extraction of immune cells to the relevant radius around each glomerulus.

Table 8 summarises the empirical computational costs associated with the construction of each of our graphs. These measurements reflect the time required and the peak memory usage observed during graph construction.

Table 8: Graph Construction Costs.

| Graph | Construction Time | Memory Peak | Nodes | Edges |
|-------|-------------------|-------------|-------|-------|
| 001 | 16.77 s | 726.70 MB | 387,746 | 2,375,072 |
| 002 | 14.62 s | 579.45 MB | 189,173 | 1,094,634 |
| 003 | 8.12 s | 500.07 MB | 73,613 | 393,892 |
| 004 | 8.69 s | 554.22 MB | 155,587 | 889,968 |
| 005 | 9.39 s | 594.59 MB | 191,592 | 1,108,688 |
| 006 | 24.64 s | 974.22 MB | 774,278 | 4,113,188 |

In addition to efficient graph construction, the usage of a GNN allows for efficient training and inference. The peak memory usage during training on whole WSIs of patients 001-003 was measured at 1943 MB, and the average training time per epoch was 1.21 s. These metrics illustrate that our method not only scales well in terms of graph construction but also during model optimisation, making it feasible for clinical applications.

All experiments were performed on a workstation equipped with an NVIDIA Titan RTX and a NVIDIA Quadro RTX 6000 GPU with 24GB of VRAM, running Ubuntu 20.04. This configuration provided a sufficient environment for both graph construction and model training, as reflected in our reported computational metrics.

## Appendix J. Ablation Study - Stain Augmentation

To test the models' robustness against image domain shifts, we applied stain augmentation to the WSIs for further experiments, as proposed by Tellez et al. (2018) to create diverse and realistic stain variations. We increased and decreased the intensity of each of the three stain signals by 50%. Table 9 presents the results of the three best-performing models tested on augmented WSIs from patients 004-006. In most cases, the modified U-Net outperforms the HIEGNet and the Random Forest. However, our HIEGNet significantly outperformed in only 2 of 6 test scenarios. This performance difference can be explained by the pre-training

of the U-Net. As discussed in Appendix D, the U-Net was pre-trained using self-supervised learning on a dataset with diverse staining protocols, while HIEGNet was trained on only three WSIs with the same staining protocol. When taking this immense difference in the training set-up into account, it can be hypothesised that with further modification in the training set-up and the amount of training data, our HIEGNet has great potential for domain transfer. We currently see this as the most promising direction for future work on the graph-based approach to histopathological data.

Table 9: F1-Scores macro-averaged for HIEGNet, the Random Forest and the modified U-Net for different stain augmentations tested on patients 004-006.

| Augmentation | Random Forest | U-Net | HIEGNet |
|---|---|---|---|
| No augmentation | 0.64±0.01 | 0.52±0.08 | **0.73±0.02** |
| DAB + 50 % | 0.43±0.02 | **0.49±0.09** | 0.42±0.02 |
| DAB - 50 % | 0.37±0.01 | **0.56±0.08** | 0.49±0.02 |
| CD3 + 50 % | 0.36±0.01 | **0.62±0.05** | 0.43±0.02 |
| CD3 - 50 % | 0.46±0.01 | 0.31±0.06 | **0.51±0.02** |
| CD68 + 50 % | 0.43±0.02 | **0.55±0.07** | 0.47±0.03 |
| CD68 - 50 % | 0.37±0.01 | **0.46±0.10** | 0.45±0.02 |

## Appendix K. Ablation Study - Edge Importance

The results presented in Section 5 demonstrate that a graph-based approach performs well in the task of glomerular health classification. Our method is based on the hypothesis that the immune environment and neighbouring glomeruli provide valuable information for this task. To assess this hypothesis, we evaluated the performance change of HIEGNet in the within patients setting after systematically removing all edges of each of the three edge-type groups: $r_{g,g}$, $\mathcal{R}_{ig}$, and $\mathcal{R}_i$. The results, presented in Table 10, show that the performance declines in all three cases. The largest decrease in F1-score, with 0.07, occurs when removing edges between immune cells and glomeruli, suggesting that these connections are the most critical for the classification task. These results support our hypothesis that the immune environment is relevant to the sclerotic state of a glomerulus and that a GNN can effectively model this phenomenon.

Table 10: F1-Scores of HIEGNet after removing edges, where the column entitled "Delta" measures the performance change with respect to the model where all edges are present.

| Removed Edge Types | F1-Score | Delta |
| :---: | :---: | :---: |
| $r_{g,g}$ | $0.69 \pm 0.02$ | -0.04 |
| $\mathcal{R}_{ig}$ | $0.65 \pm 0.02$ | -0.07 |
| $\mathcal{R}_i$ | $0.70 \pm 0.02$ | -0.03 |

