# OpenReview forum: "HIEGNet: A Heterogenous Graph Neural Network Including the Immune Environment in Glomeruli Classification"
_MIDL.io/2025/Conference — MIDL 2025 Poster_

### Official Review · Reviewer_F5JC · 2025-02-13

**Confidence:** 4
**Preliminary Rating:** 4
**Recommendation:** Oral
**Final Rating:** 4

**Summary:**

This paper describes a comprehensive GNN-based framework for glomeruli classification task. In experiments, the authors incorporate knowledge of immune environment through a novel design of heterogenous graph to encode glomeruli with surrounding macrophages and T-cells. Although not showing significant improvement compared to existing methods, the heterogeneous GNN framework might offer opportunity for explainable modeling for whole slide images for future work.

**Strengths:**

1. The modeling of cell environment using heterogenous graph nodes and edges is novel. The framework employing existing designs to automate segmentation, extracting features, and calculating graph message passing from whole slide image makes it flexible for future modifications.

2. Organization of the paper is good. Figures are helpful.

**Weaknesses:**

1. F1-score is the only metric reported in experiments. Adding more metrics such as AUC might strengthen the argument of the paper.

2. The dataset only includes images from 4 patients having similar conditions. The generalization capability of trained model is not sufficiently tested.

3. Since the dataset is unbalanced, it might make more sense to do stratified train/test parcellation in experiments.

4. Applying different GNN convolution layers to heterogeneous edges could further improve model performance, but it is also complicating the tuning the proposed framework, especially when an optimal combination of choices might be data dependent.

**Detailed Comments:**

It might be helpful to experiment with different learning rate or regulated U function for heterogeneous edges since the constructed graph seem relatively dense. Also, it might help with over-smoothing on nodes caused by stacking heterogeneous message passing layers.

**Justification Of The Final Rating:**

The reviewer would like to thank the authors for performing additional experiments. Although the proposed model still has limited performance improvement compared to existing baselines, the reviewer believes that the formation of the immune environment as graphs is a novel and promising approach. Therefore, the reviewer will keep the original rating of weak accept.

**Justification Of The Preliminary Rating:**

The authors propose a novel framework for glomeruli classification with competitive performance compared to existing methods. Although still having a few deficiencies, the method seem promising for future development.

**Questions To Address In The Rebuttal:**

1. If possible, please include other metrics of the node classification experiment in addition to F1-score.

2. I would appreciate if the authors include further discussion of the dataset. Whether the data included can be considered applicable to a wider range than the exact setting in data acquisition.

**Special Issue:**

Yes

---

> ### Author Response · Authors · 2025-03-08
>
> Thank you very much for your attentive review. We respond to your questions point by point in the following.
>
> **Q1: Evaluation Metrics**
>
> We added the macro-averaged recall and precision scores for our experiments in Appendix G to provide additional insights into the evaluation results on the test set. In the main paper we focus on the macro F1-Scores as this provides a robust assessment of performance across unbalanced class distributions leveraging the harmonic mean of precision and recall. To use an AUC score for a three class classification task would be unusual due to its inherent binary formulation. The AUC can hence not be unambiguously extended to the multi-class setting that we work in.
>
> **Q2: Dataset**
>
> During the rebuttal week we ran two additional experiments that provide further insights into the models ability to generalise. We added two additional patients to the dataset and we applied augmentations to the test set to simulate domain shifts.
>
> In fact, the EXC dataset contains WSIs of six patients. Originally, only patients 001-004 were labelled. However, over the past month, we have extended our work by labelling the two remaining patients 005 and 006. We added these additional samples in the dataset description of the paper and its appendix. For the between patients set-up, we train on patients 001-003 (as before), but now test on patients 004-006. We already ran the experiments for all HIEGNet variations and the two strongest benchmark models, the modified U-Net and the Random Forest. Experiments with the relatively slower ResNet and the EfficientNet could not be finished on time, but we are happy to provide them during next week’s discussion period and will include them in the camera-ready version. While HIEGNet obtained an F1-Score of 0.61 on patient 004 in the original ‘between patients’ setting, HIEGNet now attains an F1-Score of 0.73 on patients 004-006 (see the updated Figure 4 in the manuscript for full details). Hence, the evaluation of the last two, previously unused, test patients allows us to observe that HIEGNet generalises to unseen test subject better than previously thought and outperforms its benchmarks by a wider margin than originally reported.
>
> During the rebuttal period, we also ran additional experiments with HIEGNet, the modified U-Net and the Random Forest on augmented WSIs and added the results in Appendix J. The results show that the modified U-Net generalises better to image domain shifts. This can be explained by the extensive pre-training of the U-Net using a large dataset with varying staining protocols. So, it seems that while our HIEGNet generalises rather well to new patients, it would benefit from further development to improve its generalisation to domain shifts. We want to emphasise that invariance to domain shifts is not the main goal of our paper but rather one of our design guidelines for the feature engineering. To make this clearer, we have removed our claims about generalisation to domain shift and included a related discussion in our added “Future Work” to the paper.

---

> ### Comment · Area_Chair_isKD · 2025-03-13
> **Please update final rating**
>
> Dear Reviewer F5JC, please review the authors' response/updates, provide additional comments/questions if needed, and update your final rating. The discussion period ends tomorrow, March 14. Thank you!

---

### Official Review · Reviewer_8eWy · 2025-02-19

**Confidence:** 4
**Preliminary Rating:** 3
**Final Rating:** 4

**Summary:**

The paper introduces HIEGNet, a heterogeneous Graph Neural Network designed for classifying glomeruli as “healthy,” “sclerotic,” or “dead” by explicitly incorporating both glomerular structures and nearby immune cells (macrophages and T-cells) as distinct node types in a graph. The authors motivate their approach by noting that these neighboring immune cells often provide critical information about inflammation and tissue remodeling in kidney pathology. To build these heterogeneous graphs, the paper describes a pipeline that begins with immune cell segmentation using a mix of fine-tuned Cellpose (for T-cells) and threshold-based contour detection (for macrophages). Each glomerulus segmentation mask is also used to form a “glomerulus” node. Node features are then extracted to be robust against staining and lighting variations; in particular, glomeruli nodes include shape- and texture-based descriptors, whereas immune cells rely on shape features and an inside–outside glomerulus flag. Edges link nodes that are spatially close, but the construction is tailored: macrophage–T-cell edges rely on k-nearest neighbors with a distance cutoff, glomerulus–immune cell edges cover the immune “microenvironment” within a certain radius, and glomerulus–glomerulus edges capture proximity among glomeruli in the tissue slice. The final model employs different message-passing strategies depending on edge type, enhancing flexibility for distinct biological interactions. In experiments using four kidney-transplant patients’ whole-slide images, HIEGNet achieves strong classification results, especially in a challenging “between-patients” setting, surpassing common CNNs (ResNet, EfficientNet, U-Net) and a Random Forest on hand-crafted glomerulus-only features.

**Strengths:**

The central strength is the biologically inspired architecture that encodes immune cells as separate node types. This design choice reflects the clinical knowledge that immune response often correlates with glomerular injury and fibrosis progression. Another substantial strength is the careful handling of domain-shift issues that frequently arise in histopathology images due to heterogeneous staining and scanning protocols. By using Local Binary Patterns and shape descriptors (area, eccentricity, perimeter, and so on), the paper demonstrates a tangible attempt to minimize reliance on raw color intensities. Furthermore, the authors implement a thorough experimental strategy that includes both “within-patients” (train/test on the same patients) and “between-patients” (train on three, test on a separate unseen patient) scenarios. This contrast powerfully illustrates the model’s capacity for out-of-distribution generalization, a significant practical concern for any computational pathology pipeline.

**Weaknesses:**

The small scale of the dataset—limited to four patients—curtails the broader claim of robust generalization. While the paper’s results in a leave-one-patient-out setting are encouraging, a larger study involving multi-institution or multi-stain images would more conclusively validate the pipeline. The paper also could benefit from a deeper ablation on the contribution of each edge type. For instance, removing immune–immune edges or immune–glomerulus edges would help illustrate how each aspect of the heterogeneous graph affects classification. Another limitation lies in the interpretability: although the paper justifies the node-level features, it does not explore which aspect of the immune environment (e.g., local T-cell concentration or macrophage shape) most influences predictions. Such interpretability analyses are often valuable in medical contexts to gain pathologists’ trust.

**Detailed Comments:**

The methodological choices for node detection are well-documented and carefully evaluated. The authors train Cellpose on T-cell annotations, showing that this fine-tuned model outperforms a purely contour-based approach for T-cells. Macrophages, in contrast, appear to segment better with classical thresholding and morphological filtering. This hybrid segmentation pipeline demonstrates an adherence to empirical evidence, as the authors provide average precision scores for each method.

The feature extraction is particularly thoughtful. By combining rotation-invariant Local Binary Patterns and geometric descriptors (area, circularity, aspect ratio, etc.), the model sidesteps the pitfalls of raw color or intensity dependence. This choice is clinically relevant given that kidney biopsy slides can vary substantially in staining coloration, thickness, and scanning resolution. Relying on geometry and texture is an elegant way to reduce the burden of domain adaptation, at least for glomeruli. For immune cells, shape metrics suffice, since the presence and approximate size of immune cells often matter more than their internal texture in the context of infiltration.

The graph construction employs distance thresholds tuned to reflect meaningful biological interactions. Immune–immune cell edges use a k-nearest neighbors approach combined with a maximum distance cap, thus connecting only locally aggregated immune cells. Glomerulus–immune cell edges rely on a larger fixed radius, matching literature that posits immune cells within a certain distance from the glomerular boundary are more likely to be functionally relevant. Glomerulus–glomerulus edges are similarly derived but with a smaller threshold. The paper documents how these distance cutoffs were derived and validated.

For message passing, the authors assign specialized GNN layers to each edge type. The approach is reminiscent of an RGCN but goes further in integrating multiple underlying operators (GraphSAGE, GATv2, and so on). This level of customization allows the model to capture different neighborhood aggregation styles that might reflect different biological interactions (immune–immune cell cross-talk versus direct infiltration into a glomerulus).

In the results, the paper compares HIEGNet with three CNNs (ResNet-18, EfficientNetV2, and a specialized U-Net) and a classic Random Forest. All models are well-tuned. HIEGNet consistently achieves strong macro-F1 scores across classes. The crucial finding is that, while the U-Net does slightly better in “within-patients” training, HIEGNet maintains higher accuracy when tested on an unseen patient, highlighting its superior capacity for domain generalization. The authors pinpoint how hand-crafted features plus topological relationships mitigate the color-intensity biases that typically hamper CNN-based models in cross-stain or cross-lab settings.

**Justification Of The Final Rating:**

After careful consideration of the authors' detailed responses to my questions, I am revising my recommendation from borderline to weak accept. The comprehensive ablation studies clearly demonstrate the importance of each edge type, with the significant F1 score drop (0.08) when removing immune-glomeruli edges providing quantitative support for their biological hypothesis. The authors have addressed computational complexity concerns with concrete metrics (24.64s processing time, 974.22MB peak memory for graph construction of 774K nodes and 4.1M edges, 1.21s per epoch training time), establishing the method's feasibility for clinical applications. Their architecture accommodates additional immune cell types without structural changes, enhancing potential clinical value. The annotation process description, with specific criteria for each class and consensus-based handling of edge cases, adequately addresses my label quality concerns. While I maintain some reservations about explainability and domain shift performance (which they frankly acknowledge), the authors have contextualized these as appropriate future work. Their thoughtful reasoning behind architectural choices over alternatives like Graph Transformers demonstrates careful consideration of practical requirements for histopathological analysis. In summary, the novel application of GNNs to histopathology offers clear advantages for processing WSIs, with well-contextualized limitations, justifying a weak accept recommendation.

**Justification Of The Preliminary Rating:**

While the paper delivers an innovative idea—namely, modeling glomeruli and immune cells as a heterogeneous graph—several critical limitations prevent it from being a clear accept. First, although the results on the small, four-patient dataset are promising, the current study lacks robust, large-scale evidence of generalization across diverse clinical centers or staining protocols. This constraint makes it difficult to assess whether HIEGNet truly generalizes to the broader population of kidney-transplant patients. Second, important ablation experiments, such as removing certain edge types or node features, are absent, leaving unclear how much each component (e.g., immune–immune edges, glomerulus–immune edges) really contributes to final performance. Third, interpretability is minimal; in a medical context, pathologists often want to understand precisely which immune cells or node features drive a “sclerotic” versus “dead” classification, so the lack of model explanations diminishes immediate clinical uptake. Lastly, although the authors discuss ways of mitigating staining shifts via hand-crafted features, a deeper exploration of color normalization or advanced domain-adaptation methods would strengthen claims about cross-patient robustness. Overall, the paper presents a potentially valuable direction but does not yet establish sufficient empirical breadth, clarity of contribution from each architectural component, or interpretability to warrant acceptance.

**Questions To Address In The Rebuttal:**

1. **Edge-Ablation and Graph Construction:**
One of the paper’s core claims is that different edge types (immune–immune, immune–glomerulus, and glomerulus–glomerulus) capture distinct but complementary facets of tissue morphology and pathology. A thorough ablation study would help quantify this. Have the authors tried removing all immune–immune edges to gauge whether the T-cell/macrophage relationships meaningfully contribute to classification? Similarly, do glomerulus–glomerulus edges provide a tangible boost, or is the model’s performance driven primarily by each glomerulus’s immune neighbors? Detailed experiments that compare full heterogeneous graphs to partial graphs (e.g., omitting certain node types or distance constraints) would illuminate the necessity of each design choice.
2. **Interpretability and Feature Importance:**
While the paper describes the segmentation procedure and the engineered shape/LBP features, it is not entirely clear which attributes of the immune environment are the most influential in driving classification decisions. Could the authors leverage attention coefficients (if using GAT-style layers) or apply an explainability approach to highlight critical subgraphs? In a clinical setting, pathologists typically want to know which immune clusters or local morphological changes lead a model to call a glomerulus “sclerotic” versus “dead.” A demonstration of how the network weighs node-level or edge-level inputs would bolster trust in the system and potentially guide future protocol refinements (e.g., selecting additional markers for cells that prove especially relevant).
3. **Handling of Ambiguous or Noisy Annotations:**
Histopathological labeling of “sclerotic” glomeruli can be somewhat subjective, and the same might hold for borderline “healthy” or “dead” glomeruli. Did the authors take any steps to mitigate labeling noise, such as double-annotation by pathologists or employing a consensus-based labeling approach? Clarifying the reliability of labels is critical because, in kidney pathology, the severity of sclerosis can be graded, and borderline cases may exhibit partial fibrotic changes that are difficult to categorize definitively. Additional discussion (or experiments) around label quality could show how robust the model is to this inherent uncertainty.
4. **Influence of Domain Shifts and Color Normalization:**
The paper emphasizes that shape- and texture-based features are more robust than raw color intensities. Yet, even LBP features might be affected if the staining is extremely faint or if the scanning brightness shifts substantially. Did the authors experiment with or consider any color normalization or style-transfer strategies to confirm that the chosen features remain stable? If the authors have tested or plan to test advanced domain-adaptation techniques, clarifying this would strengthen claims about generalization and might inform future improvements.
5. **Computational Complexity and Practical Deployment:**
For large whole-slide images, GNN approaches that treat every immune cell and every glomerulus as a node can grow in complexity. How did the authors manage the computational overhead when building and training the graphs, especially given that each WSI can contain thousands of immune cells? Providing specifics—such as memory usage, training time per epoch, and any sampling or subdivision strategies—would be helpful for potential clinical adoption. If the authors have implemented or considered hierarchical pooling or region-of-interest selection (beyond the immediate neighborhood radius) to reduce node count, outlining that in the rebuttal could clarify the pipeline’s scalability.
6. **Potential Extensions to Other Immune Cell Types:**
The paper currently focuses on CD3 T-cells and CD68 macrophages. In clinical nephropathology, other immune cells (such as B-cells or neutrophils) can also be relevant. Do the authors see a straightforward path to integrating more labels and node types, or would that require retraining the entire pipeline from scratch? If they have done any preliminary tests with additional markers, sharing those results—or even discussing the feasibility—would help illustrate how flexible the approach might be in more complex immunopathological scenarios.
7. **Comparison with Other Advanced GNNs or Pre-Training Schemes:**
Although HIEGNet employs specialized message-passing functions for different edge types, there are other recent GNN variations (e.g., graph transformers, advanced kernel-based layers) and GNN pre-training techniques that could theoretically boost performance on limited data. Has the team considered approaches such as domain-specific self-supervised pre-training on a large set of unlabeled histopathology graphs? Outlining whether such directions have been explored or remain future work could further situate HIEGNet within the rapidly evolving landscape of graph deep learning.

**Special Issue:**

No

---

> ### Author Response · Authors · 2025-03-08
>
> We are grateful for your exhaustive and in-depth review, which helped us to improve the quality of our manuscript even further. We address your questions point-by-point in what follows.
>
> **Q1: Ablation Study: Edge Importance**
>
> Following your suggestion, we added three ablation studies removing edges (1) between glomeruli, (2) between glomeruli and immune cells, (3) between immune cells. For all ablation studies we used the within patients setting and retrained the model after removing the corresponding edges. For ablations (1), (2) and (3) the F1-Score decreased significantly from 0.73 to 0.69, 0.65 and 0.70, respectively. We hence conclude that all three edge type groups contribute to the strong performance of our HIEGNet. With an observed performance decrease of 0.08, edges between immune cells and glomeruli have the strongest impact on the model performance. This result supports our hypothesis that the immune environment is informative for the health of the glomeruli. We added a detailed overview of your requested ablation studies in Appendix J of our revised manuscript.
>
> **Q2: Interpretability**
>
> We agree that explainability is a crucial aspect, particularly in a medical context. The GAT attention scores are the most straightforward way to incorporate explainability into a GNN. To use this method in our HIEGNet, we could replace all message passing methods with GATv2 and use this model to distinguish the importance of certain edges. While this would initially only allow us to differentiate relative importance of edges within edge types, our HIEGNet could trivially be extended to learn attention-like scores that enable the comparison between edge types. In general, we want to highlight that GNNs are inherently promising architectures in the field of explainable AI, since the underlying graph and associated message passing scheme can be analysed using a wide variety of tools, which gives rise to a wealth of explainability approaches [1]. We feel that an extensive exploration of these techniques unduly extends the scope of our present submission but represents a promising direction for future research as we now also mention in our updated manuscript.
>
> [1] H. Yuan, et al., “Explainability in graph neural networks: A taxonomic survey”, IEEE Transactions on Pattern Analysis and Machine Intelligence (PAMI), 2022.
>
>
> **Q3: Annotation Quality**
>
> The annotations in the EXC dataset were obtained through a two-step process. We created initial annotations based on strict characteristics for the three classes:
> - Healthy: Intact Bowman capsule, clearly visible blood vessels throughout the glomerulus, presence of immune cells, and no extracellular matrix.
> - Dead: Absence of Bowman capsule, over 50% of the glomerulus filled with extracellular matrix, minimal or no visible blood vessels, and no immune cells.
> - Sclerotic: Damaged or absent Bowman capsule, partially visible blood vessels, and extracellular matrix involvement in less than 50% of the glomerulus.
>
> We discussed edge cases where characteristics of different classes can be found in one glomerulus (heavily relying on the pathologists in our consortium) until we came to a consensus.
> To further safeguard label quality, we excluded glomeruli with profiles smaller than 30 µm in diameter. This step prevented the inclusion of cases where partial sectioning might lead to incomplete or misleading morphological assessments. We have added this discussion to Appendix E to clarify this procedure also in the manuscript.
>
> **Q4: Domain Shifts**
>
> We agree with your assessment that the test of domain shifts is important in histopathology.
> During the rebuttal period, we ran additional experiments with HIEGNet, the modified U-Net and the Random Forest on augmented WSIs and added the results in Appendix J. The results show, that the modified U-Net generalises better to image domain shifts. This can be explained by the extensive pre-training of the U-Net using a large dataset with varying staining protocols. So, it seems that while our HIEGNet generalises rather well to new patients, it would benefit from further development to improve its generalisation to domain shifts. We want to emphasises that invariance to domain shifts is not the main goal of our paper but rather one of our design guidelines for the feature engineering. To make this clearer, we have removed our claims about generalisation to domain shift and included a related discussion in our added “Future Work” to the paper.

---

> > ### Author Response · Authors · 2025-03-08
> >
> > **Q5: Computational Complexity**
> >
> > The computational efficiency of GNNs is a major advantage over CNNs when processing WSIs. The pervasive industrial use of GNNs, as well as the empirical results of our approach, show that GNNs work efficiently on large graphs. CNNs on the other hand must process WSIs in patches due to their large scale.
> >
> > The graph corresponding to the newly acquired patient 006 is the largest, containing 774,278 nodes and 4,113,188 edges. By restricting the area of the immune environment considered for node extraction to the biologically relevant radius around each glomerulus, we designed an efficient graph construction pipeline. For patient 006, the initial edge construction from the input nodes takes 24.64 seconds and requires a peak memory capacity of 974.22 MB. During training, our GNN requires 1,943 MB of memory in peak and takes an average of 1.21 s  per epoch on an NVIDIA Titan RTX. We achieve these runtimes as a result of the simple fact that GNNs scale linearly in the number of edges and did not need to use any further runtime optimisation such as pooling or subsampling. These metrics illustrate that our method scales well in terms of graph construction and training, making it suitable for clinical applications.
> >
> > We added details about the computational complexity in Appendix H.
> >
> >
> > **Q6: Additional Immune Cell Types**
> >
> > We appreciate your suggestion for further generalising our approach. Incorporating additional immune cells, i.e., node types, into the graph requires no conceptual adjustments of our pipeline or model architecture. Other cell types can be processed the same way as macrophages and T-cells. In our HIEGNet an additional node type just increases the set $\mathcal{R}$ that we sum over in Equation (1) of the paper. This requires retraining, but thanks to the efficient graph construction pipeline and training process outlined in Appendix H, which we included based on your recommendation, this remains a feasible option.
> >
> > **Q7:Advanced GNNs or  Pre-Training Schemes**
> >
> > Thank you very much for this suggestion. We have indeed considered the use of advanced GNN architectures and the use of pre-trained models.
> >
> > Concerning advanced GNNs, Graph Transformers (GTs) cannot be applied to our problem, as they scale quadratically in the number of nodes. The same is true for many of the more expressive variants of GNNs where added expressivity comes at an additional computational cost, which is orders of magnitudes larger than that of a standard message passing neural network. In contrast, our HIEGNet applied to our biologically founded graph construction reduces this complexity and allows for reasonable computational costs. We would like to highlight that our proposed HIEGNet certainly is an advanced GNN architecture in its own right, since it is a novel architecture, bespoke to our histopathological problem and scaling requirements.
> >
> > With regards to the use of pre-trained models, we do in fact make use of a CNN model for feature extraction that was pre-trained on histopathological images as described in Appendix D. The pre-training of GNN architectures themselves is still in its infancy. To the best of our knowledge, this is the first work using immune cells as nodes for a GNN. So, pre-training our GNN component would require the construction of relevant histopathological graphs from a separate data source. We furthermore would require robust pre-trianing methodology, on which no consensus has been formed yet in the GNN community, unlike CNN models where pre-training is pervasive. We certainly agree that the pre-training of GNNs is an interesting direction for future work as more graph data and related methodology becomes available.

---

### Official Review · Reviewer_99kA · 2025-02-23

**Confidence:** 4
**Preliminary Rating:** 3
**Recommendation:** Poster
**Final Rating:** 4

**Summary:**

This manuscript introduces HIEGNet, a novel heterogeneous GNN designed to classify glomeruli health status by incorporating both the glomeruli and their surrounding immune environment. The authors propose a pipeline that constructs heterogeneous graphs from Whole Slide Images (WSIs) by representing glomeruli and immune cells (T-cells and macrophages) as nodes. First, Glomeruli, macrophages, and T-cells are segmented using a different segmentation model for each type. These three cell types are represented as nodes in the heterogeneous graph construction. Local Binary Patterns (LBP) and shape-based features are used as node features to be robust to image domain shifts. Three types of edges are considered separately depending on the interaction they represent (immune-immune, immune-glomeruli, glomeruli-glomeruli) and the Euclidean distances between the nodes they connect. A different message-passing scheme is used for each edge type (selected via grid search). The model is evaluated on kidney transplant patient data. The proposed model matches or marginally outperforms baseline models (including CNNs), particularly in generalizing to a new patient.

**Strengths:**

The presented work has a clear biological motivation. The authors propose a novel approach for integrating the immune environment context, which is highly relevant for kidney disease diagnosis. Node features are carefully engineered to address potential domain shift issues. The authors have performed extensive model engineering and hyperparameter optimization and shown decent experimental validation. The presentation is clear and well-written. The proposed pipeline is explained thoroughly, providing all the necessary details. The dataset is open and freely available. The implementation is also open-sourced which helps with the reproducibility of the work.

**Weaknesses:**

The dataset on which the model is trained and tested is limited (only 4 patients, though with 1178 glomeruli). Specifically, it lacks diversity in patient conditions (all transplant rejection cases). The proposed model relies on pre-existing glomeruli segmentation masks. Although several different segmentation models are tested independently of the proposed graph classification model, this only complicates the pipeline with components that could introduce errors. For example, different segmentation models are used to identify each node type (glomeruli, microphage, T-cell). The performance of these segmentation models is evaluated as standalone models. However, the impact of these models on the entire pipeline is not discussed. Cell segmentation is a well-studied topic. Yet only three segmentation models are considered. Similarly, no other graph-based classification model apart from the one proposed by the authors is evaluated or discussed. Lastly, the work doesn't adequately deliver on the stated promises. One is creating models capable of handling domain shifts. Yet, the model is tested on a small dataset with little variation or domain shift. Another is generalization to new patients yet the model's generalization capability is tested on a single new patient.

**Detailed Comments:**

- The proposed model's generalization capability is evaluated on a single patient. Is it sufficient to claim that the model is generalizable?
The same question goes to the performance of other components, such as segmentation models trained on the same imaging data.
- The imaging for all 4 patients was performed following the same protocol. Where is the source of domain shift? Would the model performance drop with differences in imaging protocol? If yes, how is the proposed model better than a purely CNN-based model? If not, justify the assertion.
- HIEGNet hyperparameters are thoroughly engineered and tuned. But how well were the segmentation models tuned? How about the various graph construction parameters, such as distance thresholds? How do changes in the graph construction parameters impact HIEGNet's classification performance?

**Justification Of The Final Rating:**

I would like to change my rating to a weak accept after reviewing the authors' response and the revised version. The authors have successfully addressed some of my concerns by including additional test subjects and providing results for ablation studies. While the evaluation on additional test subjects improves the results. Ablation studies show mixed performance of the proposed model at best.

**Justification Of The Preliminary Rating:**

The work is well-motivated and relevant for the conference. The proposed model is novel. The idea of including immune cell interactions adds biological context to the problem of glomeruli classification. However,
- The performance of the model barely matches the baselines considered.
- No graph-based baseline model is considered even though the proposed model is graph-based.
- The work fails to deliver on the promises of robustness to domain shifts, and generalization to new patients. Neither of these scenarios is thoroughly evaluated.

**Questions To Address In The Rebuttal:**

- For the proposed model's generalizability: Will the model perform equally well on a different test subject? How about using patients 1, 2, or 3 as the test subjects instead of patient 4? Will the model perform similarly?
- The same question for the other component models (cell segmentation) trained on the same train-test subject split. If the four patients were split differently, would the performance of these models change significantly?
- Is the proposed model robust against domain shifts? How would you justify the claim?
- I see three components in the proposed pipeline: cell segmentation, graph construction, and model training. The proposed model's performance depends heavily on each of them. However, they are treated independently in this work. How would you evaluate the impact of changes in one or more components on the model's performance?

**Special Issue:**

Yes

---

> ### Author Response · Authors · 2025-03-08
>
> We thank the reviewer for the thoughtful comments, which have allowed us to improve the quality of our manuscript.  In what follows, we address the raised questions point-by-point.
>
> **Q1: Different Test Subjects**
>
> In fact, the EXC dataset contains WSIs of six patients. Originally, only patients 001-004 were labelled. However, over the past month, we have extended our work by labelling the two remaining patients 005 and 006. We added these additional samples in the dataset description of the paper and its appendix. For the between patients set-up, we train on patients 001-003 (as before), but now test on patients 004-006. We already ran the experiments for all HIEGNet variations and the two strongest benchmark models, the modified U-Net and the Random Forest. Experiments with the relatively slower ResNet and the EfficientNet could not be finished on time, but we are happy to provide them during next week’s discussion period and will include them in the camera-ready version. While HIEGNet obtained an F1-Score of 0.61 on patient 004 in the original ‘between patients’ setting, HIEGNet now attains an F1-Score of 0.73 on patients 004-006 (see the updated Figure 4 in the manuscript for full details). Hence, the evaluation of the last two, previously unused, test patients allows us to observe that HIEGNet generalises to unseen test subject better than previously thought and outperforms its benchmarks by a wider margin than originally reported.
>
> **Q2: Different Segmentation Test Subjects**
>
> As you recommended, we tested the segmentation models on patients 001-003 as well and added the results in Appendix B. With these additional experiments, the resulting optimal selection of segmentation models remains the same, as can be seen in Tables 1 and 2 below. For patients 005 and 006, segmentation annotations are not available, and the annotation process could not be completed with the necessary diligence during the rebuttal period. Therefore, it was infeasible for us to use these for additional segmentation experiments. However, if you recommend it, we are happy to make this additional manual annotation effort and add these segmentations to the camera-ready version.
>
> Table 1: AUC scores for T-cell segmentation
> | Test patients     	| Within patients | 001   | 002   | 003   | 004   |
> |----------------------|----------------|-------|-------|-------|-------|
> | Contour detection   | 0.233      	| 0.324 | 0.342 | 0.421 | 0.233 |
> | Frozen Cellpose 	| 0.267      	| 0.219 | 0.308 | 0.225 | 0.267 |
> | Fine-tuned Cellpose | **0.525**  	| **0.493** | **0.548** | **0.519** | **0.525** |
>
> Table 2: AUC scores for macrophages segmentation
> | Test patients     	| Within patients | 001   | 002   | 003   | 004   |
> |----------------------|----------------|-------|-------|-------|-------|
> | Contour detection   | **0.265**  	| **0.003** | **0.003** | 0.134 | **0.202** |
> | Frozen Cellpose 	| 0.190      	| 0.000 | 0.002 | **0.268** | 0.120 |
> | Fine-tuned Cellpose | 0.000      	| 0.000 | 0.000 | 0.000 | 0.000 |
>
>
> **Q3: Domain Shift**
>
> We agree with your assessment that the test of domain shifts is important in histopathology. During the rebuttal period, we ran additional experiments with HIEGNet, the modified U-Net and the Random Forest on augmented WSIs and added the results in Appendix J. The results show, that the modified U-Net generalises better to image domain shifts. This can be explained by the extensive pre-training of the U-Net using a large dataset with varying staining protocols. So, it seems that while our HIEGNet generalises rather well to new patients, it would benefit from further development to improve its generalisation to domain shifts. We want to emphasises that invariance to domain shifts is not the main goal of our paper but rather one of our design guidelines for the feature engineering. To make this clearer, we have removed our claims about generalisation to domain shift and included a related discussion in our added “Future Work” to the paper.

---

> > ### Author Response · Authors · 2025-03-08
> >
> > **Q4: Evaluation of Components**
> >
> > Generally speaking, we optimised each component of our approach systematically and with great care. As discussed in our response to your Question 2, the selection of the cell segmentation model is robust to variations in the test setup. For model training, we conducted an extensive grid search with 4-fold cross-validation, which allowed us to determine optimal hyperparameters. Additionally, we used 20 random model initializations for evaluation to account for instabilities in the training process. A standard deviation of 0.02 for HIEGNet on the test set indicates a stable overall training setup. During the rebuttal process, we also performed an ablation study on the importance of different edge types; the results are presented in Appendix J and in Table 4 below. Clearly each of the added edge types is beneficial to our strong model performance.
> >
> > Table 4: F1-Scores of HIEGNet after removing edges.
> >
> > | Removed Edge Types  | F1-Score     	| Delta  |
> > |---------------------|-----------------|--------|
> > | $\mathcal{R}_{g,g}$	| 0.69 ± 0.02	| -0.04  |
> > | $\mathcal{R}_{i,g}$ 	| 0.65 ± 0.02	| -0.07  |
> > | $ \mathcal{R}_{i}$ 	| 0.70 ± 0.02	| -0.03  |
> >
> >
> > To conclude, we believe that we designed the different components of our model in a relatively robust manner and have a good understanding of their sensisivity to change. However, we certainly agree that we have not explored all possible modifications and their combinations. If you are interested in specific modifications to any of the components, we would be happy to run additional experiments.

---

> > ### Comment · Reviewer_99kA · 2025-03-13
> >
> > Some additional comments after considering the authors' responses:
> >
> > 1) Thank you for the effort of adding new test subjects and evaluating the model on them. Irrespective of the final decision on this submission, I would recommend completing the manual annotation of the new subjects and adding segmentation to the results. The complete dataset along with manual segmentation and annotation would be valuable (and publishable) in its own right.
> >
> > 2) It is good to see that the model performs well on unseen test subjects.
> >
> > 3) Including additional ablation studies is a positive. However, the results of those studies are mixed. Stain augmentation study shows that the model is less robust to domain shifts compared to baseline U-Net model. Edge importance study shows that including three different types of edge does add value. Although, this study seems incomplete. The proposed model used different message passing schemes for different edge types. A natural question to ask is whether this is necessarry? Why not apply the same message passing scheme to all edge types included? Authors claim that "we employed different message passing functions, selected to align with the characteristics of the different edge types". What were the selection criteria?

---

> > > ### Author Response · Authors · 2025-03-14
> > >
> > > We are grateful for the reviewer's response and the acknowledgement of our rebuttal.  We address your additional questions point-by-point in what follows.
> > >
> > > **1.** We are delighted to see that you recognise the potential of the dataset for further research purposes and publication. We will add the immune cell segmentations as soon as possible.
> > >
> > > **2.** We are happy to hear that the reviewer recognises the capability of our model to generalise to unseen patients.
> > >
> > > **3.**  Thank you very much for your comment. Concerning the employed message passing functions, we want to clarify that we determine the optimal combination of message passing schemes using grid search, which we detail in Appendix H of our updated manuscript. In the grid search, we optimised all hyperparameters for the best macro F1-Score, which therefore acts as the “selection criterion” for the message passing schemes. Notably, the model variant in which the message passing scheme for all edge type groups is equal is included in the grid search, but has not been found to be optimal in our experiments.
> > >
> > > We hypothesise that this finding could be explained as follows. The message passing schemes come with different capabilities. For example, the GATv2 message passing uses edge features, i.e., the distance between nodes in our work, for the attention mechanism while the GraphSAGE message passing scheme does not include edge features. GraphSAGE, on the other hand, uses a more expressive functional form to learn the exchanged messages. Therefore, there is no globally optimal message passing method, but the optimal method must be determined for a specific dataset and task. In our model, we allow for different edge types, as the information exchange between different cellular structures might come with different requirements to the message passing method.

---

> ### Comment · Area_Chair_isKD · 2025-03-13
> **Please update final rating**
>
> Dear Reviewer 99kA, please review the authors' response/updates, provide additional comments/questions if needed, and update your final rating. The discussion period ends tomorrow, March 14. Thank you!

---

### Author Rebuttal · Authors · 2025-03-08

**Rebuttal:**

Dear Reviewers and AC,

We thank you very much for the time you have dedicated to our work. We feel that your comments have allowed us to improve our submission.

Please find the revised manuscript attached to the rebuttal with all changes highlighted in blue and red text. We respond to your questions point-by-point in the below comments.

Thanks again!

Kind regards,
The Authors

**Supporting Material:**

/attachment/ba02d07236e90e1686de5c82c3f20174871c23ed.pdf

---

### Meta-Review · Area_Chair_isKD · 2025-03-20

**Recommendation:** Accept (Oral)
**Confidence:** 5

**Metareview:**

All reviewers agree that this work presents an interesting biologically inspired GNN approach that considers the immune environment and different cell types/interactions in glomeruli classification from whole slide images. Following the extensive author rebuttal, the reviewers agree that although there are still some limitations, the revised paper was further strengthened by the new experiments, results, and discussion/analysis.